# Improving mental ill-health with psycho-social group interventions in South Asia–A scoping review using a realist lens

**Kaaren Mathias** [1,2] *, **Sumeet Jain** [3], **Robert Fraser** [4], **Meghan Davis** [5], **Rita Kimijima–Dennemeyer** [6], **Pooja Pillai** [1], **Smita N. Deshpande** [7], **Maria Wolters** [8]

**1** Herbertpur Christian Hospital, Emmanuel Hospital Association, Uttarakhand, India, **2** Faculty of Health, Te Kaupeka Oranga, University of Canterbury, Christchurch, New Zealand, **3** School of Social and Political Science, University of Edinburgh, Edinburgh, United Kingdom, **4** Mental Health Social Worker, Exeter, United Kingdom, **5** Department of Psychiatry, University of Oxford, Oxford, United Kingdom, **6** Independent Researcher, Leuven, Belgium, **7** Department of Psychiatry, St John's National Academy of Medical Sciences, Bengaluru, India, **8** School of Informatics, University of Edinburgh, Edinburgh, United Kingdom

* kaaren@eha-health.org

## Abstract

This scoping review aimed to synthesise current evidence related to psycho-social groups as part of community-based mental health interventions in South Asia. We used a realist lens to pay attention to the contexts and mechanisms supporting positive outcomes. We included studies published from January 2007 to February 2022 that: were based in communities in South Asia, included a group component, reported on interventions with a clear psychosocial component, targeted adults and were implemented by lay community health workers. Two reviewers extracted data on intervention components, groups and facilitators, participant demographics and enabling contexts, mechanisms and outcomes. Expert reference panels including people with lived experience of psycho-social disability, mental health professionals and policy makers confirmed the validity and relevance of initial review findings. The review examined 15 interventions represented by 42 papers. Only four interventions were solely psycho-social and nearly all included psychoeducation and economic support. Only 8 of the 46 quantitative outcome measures used were developed in South Asia. In a context of social exclusion and limited autonomy for people with psychosocial disability, psychosocial support groups triggered five key mechanisms. Trusted relationships undergirded all mechanisms, and provided a sense of inclusion, social support and of being able to manage mental distress due to improved skills and knowledge. Over time group members felt a sense of belonging and collective strength meaning they were better able to advocate for their own well-being and address upstream social health determinants. This led to outcomes of improved mental health and social participation across the realms of intrapersonal, interpersonal and community. Psychosocial groups merit greater attention as an active ingredient in community interventions and also as an effective, relevant, acceptable and scalable platform that can promote and increase mental health in communities, through facilitation by lay community health workers.

**Data Availability Statement:** Data is available on Figshare with the following DOI: https://doi.org/10.6084/m9.figshare.22123943.

**Funding:** This study was supported by Global Challenges Research Fund Investment 2018-9 via the University of Edinburgh (SJ, MW and KM). This grant supported the salary of MD. No additional external funding was received for this study. The funder had no role in study design, data collection and analysis, decision to publish, or preparation of the manuscript.

**Competing interests:** The authors have declared that no competing interests exist.

# Introduction

Mental ill-health is experienced by more than 1 in 10 people in South Asia [1], home to 1.8 billion people and over one-fifth of the world's population. Globally mental health problems are a leading cause of disease burden [2], yet 60% or more of people with mental distress in South Asia have little or no access to formal mental health services [3]. The reasons for this are complex: firstly, mental health services are patchy, poorly implemented and not integrated with primary care services [4–6]; secondly, mental health problems are stigmatised, reducing help-seeking [7, 8]; thirdly, mental and neurological problems are often culturally framed and yet too often health services are not responsive to local contexts and frameworks [9]; and fourthly, limited mental health literacy means people do not know where to start to seek help [10, 11].

There is growing evidence supporting the value of psycho-social interventions in general to address mental ill-health in low- and middle-income settings (LMIS) [12, 13], as well as evidence suggesting that group interventions specifically improve social networks and mental health [14]. Psycho-social group interventions are defined in this study as structured cognitive, behavioural and social interventions intended to improve mental health implemented among a group of people who meet together on multiple occasions [15, 16]. Supplementary research suggests groups can do this by providing a platform for rehearsal of social skills, increasing social connectedness and peer-friendship [17, 18]. Importantly, groups offer opportunities for scaling, which could address the care gap for mental health in low-income and middle-income countries [19]. They also offer a way to raise mental health awareness within communities which is particularly important in South Asian settings where social stigma limits access to mental health services [18, 20, 21]. Further, groups can improve social inclusion, discrimination and access to economic resources and allow group members to discuss and develop their own solutions [22]. Psycho-social groups are therefore a promising low-cost alternative for delivering basics of mental health care for people with mental distress in the context of South Asia, where government public health resources are limited.

Groups are described as a component of interventions in many studies from LMIS [23–25]. A recent meta-analysis of studies primarily set in South Asia found that involvement in women's groups practising participatory learning and action significantly reduced neonatal mortality [26]. However, despite research suggesting that groups, both formal and informal, can protect against mental health problems [27], they have not been evaluated independently as an intervention in the sphere of global mental health.

One of the most consistent critiques of global mental health solutions is that they propose a one-size-fits-all approach, for example, using diagnostic categories built on Western constructs [28]. Further, they do not take sufficient account of the very diverse cultural and social contexts which impact the ways that people with mental distress present, where they seek help, how they participate and how they engage with care and service provision [28, 29] leading to a mismatch in help-seeking and responses [6, 30]. An evidence review with focus on a regional area therefore has the potential to ensure that synthesis of evidence produces findings with greater validity and relevance. Additionally, there is growing attention to the relevance of context and mechanisms in evidence synthesis [31, 32] which is particularly a focus of realist review, which seek to provide an explanatory analysis of how and why complex social interventions work (mechanisms) in particular settings or contexts [33].

This review was undertaken with specific focus on the South Asian geographical area as defined by the members of the South Asian Association for Regional Cooperation, which includes the countries of India, Nepal, Pakistan, Bangladesh, Sri Lanka, Maldives, Afghanistan and Bhutan. We recognise that this region shares many of the same economic, political, cultural and social features, for example, that the majority of elderly live with or are supported by

their children, that divorce and out-of-wedlock childbearing are relatively rare [34], that fertility is declining while age at marriage is rising, and that the majority of people hold a strong religious identity [35]. Economically all these countries are classified as low- and middle-income countries (LMIC) by the World Bank [36]. Therefore, despite Iyer proposing the lack of a cohesive South Asian identity [37], there are shared cultural and economic identities that justify analysing the region as a whole [38]. A focus on a geographic region that shares broad social and economic contexts, using a realist lens, we believe increases the relevance of this scoping review of evidence related to psycho-social group interventions.

Scoping reviews seek to map the existing literature on a subject and are useful when a body of literature has not yet been comprehensively reviewed. They are particularly useful for bringing together literature in disciplines with emerging evidence, as they are suited to addressing questions beyond those related to the effectiveness or experience of an intervention [39]. The aim of this study is to synthesise current evidence related to use of psycho-social groups as part of community-based mental health interventions in South Asia using a realist lens with specific research questions as follows:

(i) What types of psycho-social, group directed mental health interventions are being delivered by community mental health workers in South Asia?

(ii) What outcomes do they deliver and how are they measured?

(iii) What are possible mechanisms that trigger positive outcomes? What constrains positive outcomes?

## Materials and methods

### Framing the scoping review

This study was motivated by a question posed by practitioners who had noted the effectiveness of group platforms to improve mental health and social inclusion in a youth resilience intervention. Benefits were noted for participants almost regardless of the content of an intervention [17]. Practitioners (KM, PP) were working with Burans, a non-profit community health initiative based in North India and administered by Herbertpur Christian Hospital, a member of the Emmanuel Hospital Association [40]. Two published studies describe the organisational practices and approaches of Burans, and describe the value of attentive and innovative community mental health practice by local non-profit providers [41, 42]. A question with a local pose and gaze means it is more likely to be relevant and to lead to practical applications [43]. The search strategy and analysis were refined iteratively, in engagement with practitioners which included representatives from both southern and northern India and also included people who are experts by experience. Findings were triangulated with reference panels of local experts, who were able to engage with findings and increase their relevance and contextual validity.

Our focus was group interventions that are targeted primarily at adults from different family groups with a component designed to accomplish at least one of the following:

- prevent or treat mental health problem (s);

- support people who live with mental health problems and their carers;

- improve resilience in the face of mental health problems.

We proposed that interventions should have a clear psychosocial component. While interventions could be short, and engage with existing groups, they should involve multiple sessions. Group interventions could also be part of larger interventions with individual, family, or screening components. We excluded interventions that were only described in popular media,

that did not describe a group intervention or where groups were members of the same family or larger than 30 people in size, that were not set in South Asia, that were mainly delivered by health care professionals or were mainly biomedical, that had a primary focus on training community health workers, that did not focus on improving psychological wellbeing or that gave insufficient detail of the intervention.

We included interventions that were assessed in different ways, from RCTs to qualitative evaluations. Where interventions were tested using an RCT, we included the Study Protocol in our data sources, since these protocols often contain important information about the intervention itself that may not fit within the word count of the final RCT report. For interventions where data were reported in several papers, we included all papers with qualitative or quantitative results.

For the purpose of this review, we define psychosocial groups as meetings of 3–20 community members who meet intentionally three times or more, with the objective of improving psycho-social wellbeing or health. These meetings should be facilitated or organised by lay community-based worker(s). Groups can act directly as interventions or be a medium for other therapeutic interventions (for example, for delivery of facilitated curriculum content), and this review considered both.

This leads us to the following detailed specification, created using the SPIDER tool [44], given that our research questions are descriptive and qualitative.

**Sample.** Adults living in the community and affected by mental health problems. We included both studies of people with mental health problems and of those who care for them. Interventions should be carried out in, and benefit citizens of, the SAARC. Interventions who targeted both adults and young people (aged 14 and above) were also included.

**Phenomenon of interest.** Psychosocial group interventions with a stated intention to support mental health in SAARC countries that are delivered by community workers or primary care health workers. Those workers should have no tertiary level training in medicine, social work, psychology, or one of the allied health professions, and they should not be training in a tertiary setting. However, these workers may be regarded as experts by their community and may have undergone rigorous apprenticeships in traditional forms of health/ medicine and physical, mental, and spiritual care provision.

A minimal description of the intervention should be available, covering who delivered it, what the content of the intervention was, and at whom the intervention was aimed.

**Design.** Study protocols, implementation studies, qualitative studies, experience reports, evaluations, case studies, randomised controlled trials

**Evaluation.** Studies should report, or, in the case of study protocols, specify quantitative or qualitative outcomes of the intervention. Reports of implemented interventions should also mention barriers to and facilitators of success.

**Research type.** mixed methods, quantitative research, qualitative research, study protocol, experience report

## Protocol development

The protocol for this review was developed iteratively. We first performed a rapid review in January 2019, using the publicly available National Institute of Health's PubMed service. The start date of 2007 was selected as the year that the Lancet launched their landmark Global mental health series [45]. We then reviewed the resulting papers to clarify inclusion and exclusion criteria; develop an easy-to-use extraction scheme within the software tool Covidence; refine the research questions; and generate seed papers for the main search. Using an initial focus on PubMed has been shown to be viable in situations where syntheses are needed urgently [46].

Once the protocol had been established, we conducted the main scoping review in July 2019, updating this in February 2022, following the steps set out by the Joanna Briggs institute [47]. Publications were reviewed by two reviewers at each stage of the process. A third reviewer mediated where there was divergence in reviewer ratings. After determining the papers to be included in the review, papers were grouped together if they reported on the same intervention in more than one paper. Data extraction and quality assessment was carried out by all authors in this paper except RMK and PP.

## Search strategy

We searched Pubmed for the initial rapid review on January 3, 2019. This was followed by searching the following electronic databases through formal queries on June 16, 2019, and then finally updated and merged with a further search on Feb 7, 2022: OVID Medline, OVID EMBASE, OVID PsycInfo, and Scopus. For all databases, we chose the most up to date version of studies, where available.

The search strategies consisted of three sets of terms, one set consisting of the names of each of the eight SAARC countries, one set for mental health, which included terms such as "mental health", "psycho-social", "mental disease", "mental wellbeing", "mental stress", "social support", and one set for the delivery mechanism, which included terms such as "community care", "community health", "task shifting", "mental health care", and "mental health service", "group". Within each of the three sets, terms were joined by „or". The search terms were chosen to ensure maximum coverage of different approaches, given that such groups are described using many different terms. In particular, we wanted to ensure coverage of interventions that follow psychosocial principles, but do not describe themselves as such. We developed the search terms based on a published mini review [48]. The three sets themselves were joined by "and". We searched for papers that contained the terms in their title, abstract, or keywords.

Google Scholar and Web of Science were used for forward citation tracking of studies that meet the inclusion criteria after full text screening. Relevant systematic reviews were identified and hand-searched for additional studies. We also hand searched the relevant WHO database for the region, IMSEAR [49].

The grey literature search was conducted in mid-2019. Searches included brain storming with an advisory group of all known community mental health organisations in South Asia, followed up by an online search (Google) for NGOs and a review of interventions listed on the Mental Health Innovation Network website [50]. A list of 37 organisations was generated through internet searches, and contact emails were located from websites and networks of the advisory group. Emails were sent to the director of each organisation outlining the study objectives and requesting organisations to send relevant documents or links to documents. This exercise yielded responses from two organisations. We also conducted face-to-face meetings with relevant community mental health experts in India and Nepal.

**Data extraction and synthesis.** We documented how the outcomes of the interventions were measured, which scales were used, and how they were reported qualitatively as well as quantitatively. We paid particular attention to potential mechanisms reported in qualitative findings or in the discussions of included studies.

We extracted study characteristics that were relevant to the context of the intervention, the type of psycho-social group mental-health intervention, information about the implementation of these groups and the composition of these groups, barriers to participation, outcomes linked to psychosocial groups, and potential mechanisms for these outcomes. Quantitative study characteristics and outcomes were summarised descriptively. Qualitative outcomes were analysed using thematic analysis, as suggested by Levac et al. [51].

**Table 1. Quality assessment.**

| Intervention | Level of detail given |
|---|---|
| | Description of community health worker selection |
| | Description of community health worker training |
| | Acknowledging challenges |
| | Sufficient information for realist synthesis |
| Quantitative Findings | Selective Reporting |
| | Missing Outcomes |
| | Reporting of non-prespecified outcomes |
| | Reporting of non-prespecified analyses |
| | Reporting of outcomes for subgroups |
| | Consistent reporting |
| | Sufficient reporting |
| Qualitative Findings | Multiple coders |
| | Quality check for codes performed |
| | Description of analysis method |
| | Correct execution of analysis method |

Typically, scoping reviews do not include appraisals of the publications reviewed. We decided to report basic appraisal information to inform the planning of further dedicated qualitative and quantitative syntheses. We created a custom tool with three main parts: Quality and depth of the intervention description; Quality of any quantitative findings [informed by the CONSORT reporting guidelines]; Quality of any qualitative findings (informed by the SRQR reporting guidelines). The tool is summarised in Table 1. This tool was applied on the level of the intervention rather than on the level of individual papers. We did this because different analyses from the same intervention provided different levels of detail, and a focus on an intervention [which be published with multiple outputs] gave us the most complete information to assess the quality of an intervention.

**Stakeholder consultation.** This review was undertaken by researchers living in South Asia and high-income countries. Recognising that the deepest knowledge about local contexts is held by communities and practitioners based in those settings [43], findings from the initial rapid review were reviewed by two reference panels who validated and prioritised preliminary findings. The first panel comprised 11 mental health practitioners from across India which included psychiatrists, social workers, policy makers, researchers, non-profit and governmental organisations and service user movement representatives (Practitioner panel). They met in Delhi on 2nd May 2019 in a workshop facilitated by KM, PP and SD. The second panel was made up of 11 experts by experience (EBE panel) i.e. people with lived experience of mental health problems residing in informal urban communities in Dehradun, North India. They met on 28th June 2019 in a workshop facilitated by KM and PP. Findings from these reference panels are threaded through the review outputs.

Ethics approval for the two panel workshops was granted by the Institutional Ethics committee of the Emmanuel Hospital Association, New Delhi in March 2019. All participants in the expert panels gave signed consent for the participation and data.

## Results

This review examines 15 interventions, which were documented in 42 papers. Only a handful of included studies provided sufficient detail on the context and mechanisms to allow formation of context–mechanism- outcome configurations generated in realist synthesis [33]; so

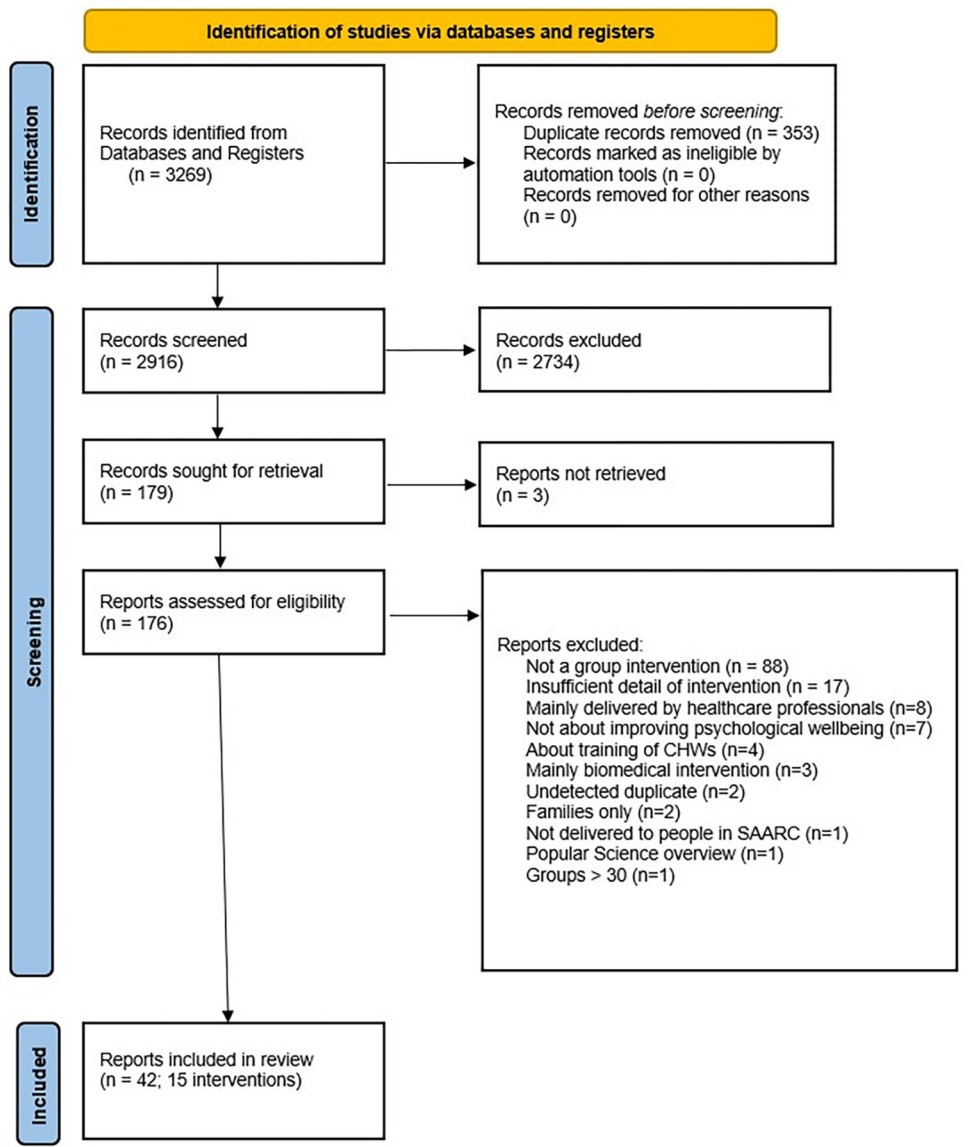

**Fig 1. PRISMA diagram showing study selection process.**

instead, we used a realist lens and examined available information on enabling contexts, and interventions to identify possible mechanisms that trigger mental health and outcomes linked to psychosocial groups. We show the study selection process in Fig 1 with a PRISMA diagram.

### Study context

All of the interventions were conducted in at least one of the SAARC countries, three of which were multi-country. The countries in which the studies took place were India (9 interventions / 17 papers), Pakistan (4 interventions / 15 papers), Bangladesh (2 interventions / 3 papers), and Nepal (3 interventions / 4 papers) and multiple South Asian sites [52]. Rural and disadvantaged communities were included in multiple studies. Seven studies were set in communities affected by a humanitarian crisis such as earthquake or tsunami [53–59].

Participants in groups were primarily women in the reproductive age range, many of whom experienced socio-economic disadvantage such as completing fewer than ten years of schooling however several studies also included men as participants [58–64]. And a few studies only included people with experience of mental distress: [18, 60–62, 64–66].

Table 2 summarises key study characteristics to show the wide range of contexts, participants and types of psychosocial groups. Detail of the demographics of study participants is provided in S1 File.

## Types of psycho-social group mental health interventions

Community mental health workers in South Asia deliver diverse group mental health interventions in a range of contexts. Each study utilized a distinct theoretical framework, and the frequency of delivery varied greatly. Some delivered sessions intensively for short periods of time, such as 60 minutes daily for eight days [56], while others met over a longer time, for example, monthly for 20 months [52, 60, 69];. Despite the heterogeneous approaches, several common themes emerged:

Most studies sought to build skills and knowledge in mental health using psychoeducational components, [19, 22, 52, 60, 61, 63–65, 68, 69, 72, 73, 77, 79, 82]. Several interventions provided both a mental and physical health component or addressed women-specific issues like perinatal depression and child health [22, 52, 68, 69].

Typically, mental health interventions combined with other forms of support. Only three studies implemented a mental health intervention alone. Eight of the 15 interventions offered economic support such as microcredit loans [55, 72, 75], emergency funds [68], and discussing strategies for generating income as part of practical problem-solving within therapy sessions [22, 55, 62, 72, 75, 78, 79]. While the other interventions reviewed did not provide tangible economic support, several emphasized socio-economic status as a key mental health determinant [18, 55, 60, 66, 72].

In addition to building economic skills, other interventions included supporting groups to participate in rights-based activities [19, 62] or promoted collective action from psychosocial groups such as advocating with local authorities to improve mental health services [62, 68, 73].

## Implementation of groups

In most studies, group facilitation was conducted by women from the community with 10–12 years of education, although a small number of facilitators were more highly trained and from outside the community [55, 61, 72]. Groups that were facilitated by local community members with lived experience shared with participants reported that this led to increased acceptability and relevance of groups [18, 52, 55, 61–63, 68, 69, 80, 82, 83, 87]. The majority of interventions provided training in group facilitation skills using a 'train the trainer' model as well as providing training in structured content [for example a curriculum] which was followed by ongoing coaching. All studies financially remunerated facilitators to a varying degree although one study reported that facilitators were also motivated by altruism and 'in kind' returns from community members [63]. Details on processes of facilitator recruitment, training and groups (duration, frequency, group size and meeting content) are also summarised in Table 2.

Groups using vignettes, roleplays and games and dialogue/ conversations found this approach was more likely to lead to engagement as they were culturally accessible and used a participatory format [52, 62, 63]. Authors also described increased compliance due to 'social prestige' in practising behaviours recommended by the group [69]. Facilitators with a good reputation and who provided training, support, feedback and sustained support (and who refrained from advice giving) were more highly regarded by group members [18, 63, 69, 87].

**Table 2. Key study characteristics.**

| Intervention title | Papers included [TOTAL] | Design | Country / setting | Group participants [based on inclusion criteria] | Group activities/ components | Group frequency | Group facilitators | Training process for facilitators |
|---|---|---|---|---|---|---|---|---|
| Widows in NE India | [TWO] Devine et al., 2007 [67] | Protocol of above | India | Widows of intravenous drug users known to local NGOs. | • Discussed mental illness, strengths, social inclusion and stigma, income generation and action plans for improving mental health.<br>• Travel and childcare costs were covered. | 10 meetings. 1x/fortnight. | Widows from each group acted as peer facilitators. | Train-the-trainer model. NGO workers and research officers were trained, and they then trained the peer facilitators. |
| | Kermode et al., 2008 [22]; | Quasi-experimental trial report of results | India. | As above | As above | As above | As above | As above |
| Service users in Gujarat, India | [ONE] Pathare et al., 2021 [19] | Pragmatic implementation trial | India. | Service users and staff in public mental health services in Gujarat, India, implementing the WHO QualityRights programme. | • Saathi [family] groups provided emotional support for carers and enabled members to participate in their relative's care<br>• Maitri [service user] groups-built peer networks of people able to support each other and participate in planning service delivery. | Not reported. | Existing healthcare professionals at each site. | Trained in recovery-oriented care and basic communication skills |
| People with mental distress in Uttarakhand, India | [ONE] Mathias et al., 2020 [62] | Prospective cohort study, uncontrolled | India | People with common or severe mental distress and people with epilepsy aged 14y and over | •Psychosocial support groups, financial inclusion, support in access to Government entitlements | 10 meetings. 1x/ fortnight | Community mental health workers | Trained 15 days in community mental health and ongoing monthly training [26 days]. Total 41 days |
| Participatory groups with mothers across South Asia | [THREE] Tripathy et al. 2010 [68] | Cluster randomized controlled trial examining participatory women's groups | India | Women aged 15–49 years who had given birth during the study period [July 2005-July 2008]. Open cohort | • Collective prioritisation of problems and solutions around maternal and neonatal health.<br>• Information about clean delivery practices and care-seeking shared using case studies and games.<br>• 10 village reps in health committees 1x/2 months | 20 meetings. 1x/month. | Local woman identified by community. | Seven days residential training course and support through fortnightly meetings with district coordinators. |
| | Clarke et al., 2014 [69] | Secondary data analysis of impact on psychosocial distress of study by Fottrell et al 2013 and Houweling et al 2011 [70, 71] | Bangladesh. | Women aged 15–49 years resident in three rural districts of Bangladesh in 810 groups | • Collective prioritisation of problems and solutions<br>• Both intervention and control clusters received health system strengthening activities<br>• Mental health issues were not explicitly addressed | 20 meetings. 1x/month. | Local women [at least some high school education]. | Seven days training with refresher training given 6 months later and frequent supervision by senior staff. |
| | Morrison et al., 2019 [52] | Qualitative analysis of all studies in this group | India, Bangladesh, Nepal and Malawi. | N/A | N/A | N/A | N/A | N/A |

*(Continued)*

**Table 2.** (Continued)

| Intervention title | Papers included [TOTAL] | Design | Country / setting | Group participants [based on inclusion criteria] | Group activities/ components | Group frequency | Group facilitators | Training process for facilitators |
|---|---|---|---|---|---|---|---|---|
| **Women in self-help groups in India** | [ONE] Rao et al., 2011 [72] | Qualitative study. | India. | Women currently part of a microcredit self-help group. | • Established savings and credit self-help group (group lending, saving, borrowing etc). • Intervention started with a song to transition o personal concerns. • Breathing exercises and discussion of problems | Ongoing. 1x/ 2weeks. | NGO workers facilitated first 10 sessions. Group members continue facilitating after this. | NGO workers trained through one week classroom-based training, pilot counselling and feedback sessions. NGO staff then trained group facilitators. |
| **Women surviving tsunami in India** | [FOUR] Becker 2006 [53] | Experience report with no control | India | People with lived experience loss of loved one in tsunami | As above | Not reported | Available NGO, teachers and government workers | Training of trainers approach– 3 days |
| | WHO India 2006 [57] | Experience report of psychosocial programme delivered post tsunami with no control | India | Community members–women and men affected by the tsunami along coastal regions of Kerala, Tamil Nadu and Andhra Pradesh | As above | Not reported | Health workers, auxiliary nurse midwives (ANMs), Anganwadi (pre-school) and others | Training of trainers approach– 3 days |
| | Becker 2007 [54] | Experience report with no control | India | People with lived experience loss of loved one in tsunami | As above | Not reported | Available NGO, teachers and government workers | Training of trainers approach– 3 days |
| | Becker, 2009 [55] | Quasi-experimental, non-randomised cluster-controlled trial. | India. | Women who live in Cuddalore and were affected by the 2004 Indian Ocean tsunami and were available to participate for study duration. | • Sharing experiences, emotional support and relaxation exercises in group sessions. • Cultural metaphors and spiritual beliefs incorporated to facilitate coping. • Individual sessions for personal and family support | Approx. 36 sessions. 3x/ week for 3 months. | Existing NGO and community health workers. | Three-day train-the-trainer model developed by 3 psychiatrists and a social worker from National Institute of Mental Health and Neurosciences [NIMHANS]. |
| **Women in North India** | [FOUR] Gailits, 2017 [73], Gailits, 2017 [74], Mathias, 2018 [42]., Gailits, 2019 [18] | Qualitative case studies of the same intervention in the same place– different research questions | India | Hindi or Garhwali speaking women impacted by mental health problems either personally or as caregiver in Dehradun, North India | Psychoeducation and psychosocial support for women with common mental health problems [anxiety and depression] | 1x/1-2 weeks for 6 + months. | Local NGO workers. | Community-based team members employed by NGO train community volunteers in mental health knowledge, referral.group facilitation and counselling skills. |
| **Women with depression in Bangladesh** | [ONE] Karasz, 2021 [75] | Pilot randomised controlled trial with small numbers [n = 24 control and n = 24 intervention] | Bangladesh | Women residing in target area with depressive symptoms and PHQ9 score >10 | Six-month group-based, fortnightly depression management and financial literacy intervention, which was followed by a cash-transfer of $186 [equivalent to the cost of two goats] at 12 months' follow-up | Fortnightly over 6 months | Trained peer facilitators from same community | Manualised training over 5 days |

(*Continued*)

**Table 2.** (Continued)

| Intervention title | Papers included [TOTAL] | Design | Country / setting | Group participants [based on inclusion criteria] | Group activities/ components | Group frequency | Group facilitators | Training process for facilitators |
|---|---|---|---|---|---|---|---|---|
| Adults in Nepal | [ONE] Jordans, 2017 [60] | Prospective cohort study. | Nepal. | Adults with a primary diagnosis of epilepsy or psychotic disorder. | Patient support groups run by community counsellors. No further details reported of the components. | Not reported. | Community counsellors | Base training in counselling plus 21-day training in psychosocial support. |
| Problem management Plus [PM+] | [SIX] Khan et al., 2019 [65] | Cluster randomized feasibility trial. | Pakistan. | Women who scored above 2 on the General Health Questionnaire [GHQ] and above 16 on the WHO Disability Assessment Schedule [WHO-DAS]. | Psychoeducation, stress management, problem management, behavioural activation and strengthening social support. | 5 sessions. Every day for a week. | Local female lay-helpers with 16 years of education and no prior experience | 6 days training by a master trainer. Followed by 4 weeks of practice cases and weekly virtual group supervision |
| | Chiumento et al, 2017 [76] | Protocol for a cluster randomised control trial. | Pakistan | Women in rural Pakistan in a humanitarian setting | • Psychoeducation, stress management, problem management, behavioural activation and strengthening social support. | 5 sessions. 3hr/session. | Local community members with 16+ years education. | 8 days training by master trainer. Weekly supervision and for supervisors, fortnightly supervision virtually. |
| | Sangraula et al., 2018 [66] | Protocol for a cluster randomised control trial | Nepal. | Adults who scored above 2 on the General Health Questionnaire [GHQ] and above 16 on the WHO Disability Assessment Schedule [WHO-DAS]. | • Psychoeducation, stress management, problem management, behavioural activation and strengthening social support. | 5 sessions. 3hr/session. | Local community members with 10+ years education. Additional group 'helpers' assisted facilitators | • 20-days training on basic psychological skills then 10-day group PM+ training. • Group 'helpers' receive basic 2-day training. • Competence/ fidelity assessed. |
| | Rahman et al., 2019 [61] | Report of cluster randomised control trial. | Pakistan. | Women who scored above 3 on the General Health Questionnaire [GHQ] and above 17 on the WHO Disability Assessment Schedule [WHO-DAS]. | Psychoeducation, stress management, problem management, behavioural activation and strengthening social support | 5 sessions. Every day for a week. | Local graduates with Bachelors degrees and no prior healthcare experience. | 7 days training by a master trainer. Weekly virtual group supervision. Supervisors supervised by master trainer. |
| | Sangraula et al., 2020 [58] | Report of feasibility cluster randomised control trial [Sangraula et al 2018 above] | Nepal | Adults who scored above 2 on the General Health Questionnaire [GHQ] and above 16 on the WHO Disability Assessment Schedule [WHO-DAS]. | • Psychoeducation, stress management, problem management, behavioural activation and strengthening social support. | 5 sessions. 3hr/session. | Local community members with 10+ years education. Additional group 'helpers' assisted facilitators | • 20-days training on basic psychological skills then 10-day group PM+ training. • Group 'helpers' receive basic 2-day training. • Competence/ fidelity assessed. |
| | Van't Hof et al., 2020 [59] | Protocol of GpPM + for adults [2-arm single blind cluster randomised control trial] | Nepal | Adults over 18 years in selected districts who score above 16 on WHO -DAS [disability assessment] and answer affirmative on the Heartmind screen | • Psychoeducation, stress management, problem management, behavioural activation and strengthening social support. | 5 sessions. 3hr/session. | Local community members with 12+ years education | • 10-days training on basic community psychological skills then 10-day group PM+ training. • Competence/ fidelity assessed. |

*(Continued)*

**Table 2.** (Continued)

| Intervention title | Papers included [TOTAL] | Design | Country / setting | Group participants [based on inclusion criteria] | Group activities/ components | Group frequency | Group facilitators | Training process for facilitators |
|---|---|---|---|---|---|---|---|---|
| **Tsunami survivors in the Andaman Islands** | [ONE] Telles et al. 2007 [56] | Pre/post comparison. | India | Survivors from the Andaman Islands of the December 2004 Indian Ocean tsunami, including both 'endogenous people' and 'mainland settlers' from India. | Yoga sessions involved exercises, regulated breathing and yoga-based guided relaxation. | 60 min/day for 8 days. | Yoga teachers trained in Vivekananda yoga system with 1-year certificate course. | N/A |
| **Social support among Pakistan women** | [ONE] Hirani et al., 2018 [77] | Randomized controlled trial. | Pakistan. | Women with no pre-existing mental health diagnosis. | • Intervention group focus on understanding stress and social support. • Control group had a one-off session about mental health | 6 sessions. 1x/week. | Local female community members. Minimum of grade 10 education | 10–15 hours 1:1 training from researchers. |
| **Economic skill building among Pakistan women** | [THREE] Hirani et al., 2010 [78] | 3-arm randomised control trial protocol | Pakistan. | Women open to local Adult Literacy Centres. | • ESB group: modules included effective communication, time management, parenting and strategies for dealing with harassment and abuse. • Counselling group: modules included stress and anger management, effective communication, active listening and supportive problem-solving. | 8 sessions. 1x/week. | Community members with at least primary education and literacy skills. | 21 hours of training from the research team. |
|  | Hirani et al, 2010 [79] | As above | Pakistan | Researchers developing economic skill building intervention | As above | As above | Researchers | NA |
|  | Asad et al, 2011 [80] | Using participatory approaches in training facilitators | Pakistan | Lady Health Workers receiving training to run economic skill building | As above | As above | Lady Health workers who hold a formal role in health information | 12 hours of training in counselling to LHWs |
| **Adults with depressive symptoms in Pakistan** | ONE Saleem et al, 2021 [64] | Open non-controlled trial | Pakistan | People with depressive or anxiety symptoms attending a primary care centre in target area, Karachi | • Group discussion on co-produced topics such as relationship management, increasing self-awareness, psycho-education | 6 sessions 1x / month | Community-based counsellors with at least 6 months experience | 1 day training in group facilitation and intervention overview |

(*Continued*)

**Table 2.** (Continued)

| Intervention title | Papers included [TOTAL] | Design | Country / setting | Group participants [based on inclusion criteria] | Group activities/ components | Group frequency | Group facilitators | Training process for facilitators |
|---|---|---|---|---|---|---|---|---|
| Thinking Healthy | [TEN] Singla, D 2014 [81] | Evaluate feasibility of peer delivery of maternal MH work | Pakistan & India | Women over 18 who were in their third trimester of pregnancy, registered local Lady Health Workers [LHWs] and scored <10 on PHQ-9 questionnaire. | • Mother's health, mother-infant relationship, importance of psychosocial support from others. • Focus on behavioural activation and challenging unhelpful thinking. | 10 individual sessions and 4 monthly group sessions. | Female community members: minimum 10 years schooling; 30–35 years old; married +children. | Train-the-trainer model. Mental health specialist supervised THPP trainers. THPP trainers then train and supervise local peer volunteers |
| | Sikander et al., 2015 [82] | Protocol for main trial | Pakistan | Depressed mothers from 3rd trimester pregnancy to 6 months post-partum | Peri-natal mental health group intervention delivered by peer facilitators with 10 individual sessions and 4 group sessions | As above | As above | As above |
| | Atif, N et al 2016 [83] | Qualitative evaluation of acceptability of peer facilitators | Pakistan | Depressed mothers and other key stakeholders like husbands and mothers in law | As per Sikander 2015 trial | As above | As above | As above |
| | Turner, E 2016 [84] | Protocol for THPP over a 36-month period | Pakistan | Depressed mothers | As per Sikander 2015 trial | As above | As above | As above |
| | Atif, N et al 2017 [85] | Adapt Thinking Healthy and feasibility study qualitative | Pakistan & India | Depressed mothers and peers | As above evaluating process and relevance of adapted TH intervention delivered by peers | As above | As above | As above |
| | Nusrat, H et al 2018 [86] | Quantitative pre-post study of Learning thru Play and TH [abstract] | Pakistan | Depressed mothers | As per Sikander 2015 trial | As above | As above but co-facilitated with a clinical psychologist | As above |
| | Atif, N et al 2019 [87] | Qualitative evaluation of model for training for peer facilitators | Pakistan | Lay peer facilitators | As per Sikander 2015 trial | As above | As above | As above |
| | Sikander et al., 2019 [63] | Results of main cluster randomised control trial. | Pakistan. | As per Sikander 2015 trial | As per Sikander 2015 trial | As above | As above | As above |
| | Ahmad et al., 2020 [88] | Evaluate implementation strength of Sikander et al 2015 trial | Pakistan | As per Sikander 2015 trial | As per Sikander 2015 trial | As per Sikander 2015 trial | As per Sikander 2015 trial | As per Sikander 2015 trial |
| | Maselko et al., 2020 [89] | Evaluate participants of Sikander 2015 trial | Pakistan | As per Sikander 2015 trial | As per Sikander 2015 trial | As per Sikander 2015 trial | As per Sikander 2015 trial | As per Sikander 2015 trial |

Both expert panels endorsed the value of trained facilitators from the community, particularly when starting a group. The EBE panel also proposed that use of dialogue rather than didactic approaches to teach knowledge and skills, were more effective at enabling positive mental health outcomes.

## Group composition

Groups that were made up of people who were socio-demographically similar were considered easier to facilitate and built trust and connection more quickly [18, 22, 52, 62] and in line with

this, nearly all studies in this review focused on groups where membership and facilitation was female. Studies described that when members shared culture, values, and beliefs this facilitated peer friendships and relationships outside the group setting [18, 64, 69] which was valuable for building long-term social support. Conversely, other studies found that groups with heterogeneous membership were more challenging to facilitate [63, 72]. One study described women who were employed or with more years of education were more rapidly able to build social connections and networks [18, 52]. Groups that functioned for more than several months reported growing trust, sharing of personal issues which led to the formation of peer friendships [18, 61, 62, 64, 72, 83, 87].

The Practitioner panel noted that most groups in this review were primarily for women, suggesting that men are less likely to join groups because they typically are away from home during work hours. They suggested that it would be useful to develop groups to meet men's psycho-social needs that are specifically facilitated by men, for men. They highlighted possible synergies could build on pre-existing community groups, such as sports or farmers' groups to meet the psycho-social needs of men.

## Barriers to participation in psycho-social groups

Joining groups was more difficult for people with a stigmatised social identity, such as widows of intravenous drug users [22] or people with severe mental illness [63, 82]. To mitigate this, one intervention provided a financial contribution to enable disadvantaged women to join programs [22]. Others, however, identified that financial incentives led to disputes and poorer group relations [18, 63, 69].

Being able to participate in groups was also a challenge for women without freedom of movement, due to the restrictive gender relations in South Asia. This was particularly difficult for women who were socio-economically disadvantaged [18, 55, 62–64, 69, 82]. In fact, gender relations were such a strong barrier to women's group participation in one study, that restrictive gender relations were identified as key to an inconclusive result [69].

Further barriers were linked to concerns about the confidentiality of personal information and cultural relevance of the intervention content. Several studies found that participants were unwilling to disclose personal information [22, 72, 87]. In several studies, group members expressed concerns about sharing in groups where group facilitators and members are often part of the same community [61, 63]. In addition, challenges were linked to culturally adapting group resources to ensure they were linguistically and contextually appropriate [22].

The Practitioner panel proposed confidentiality concerns as an important barrier to group participation, and suggested this is more challenging where a group is led by a community peer. The EBE panel suggested strategies to mitigate this include raising and discussing confidentiality issues at an early stage, such as including a statement on confidentiality in collectively developing group ground rules.

## Outcomes linked to psychosocial groups

Studies in this review used 46 different validated quantitative measures, 38 [83%] of which were built on Western biomedical constructs and just eight measures developed in South Asia. Psychometric measures can be less valid or accurate when they are used in a different setting or country to the place where they were originally developed. In a new setting they may fail to describe or represent the features of mental health or sickness which are determined culturally, socially and politically [90]. There is an 'ongoing need for in the conceptualization and measurement of culture- specific psychopathology and in developing culturally responsive interventions' [91]. Outcomes were not easily comparable because of the range of interventions,

**Table 3. Variables and scales used for social and health outcomes in this review.**

| Type of scale | Scale |
|---|---|
| Mental Health/ psychological [n = 19] | Generalised Health Questionnaire (GHQ-12), Hospitalised Anxiety and Depression Scale (HADS), Kessler (K-10), xPTSD criteria for DSM-5, Impact event score, Patient Health Questionnaire (PHQ-9), Positive and negative symptoms scale, Hamilton Depression Scale, Epilepsy questionnaire, 10-point scales on fear, anxiety, disturbed sleep and sadness, Beck Depression Inventory II, Trait hope scale, Post-traumatic stress disorder checklist (PCL-5), Self-reporting Questionnaire (SRQ-20), Strengths and difficulties (SDQ-TD), Aga Khan University Depression and Anxiety Scale*, Psychosocial mental health problems (PMHP scale)*, PaCoMSI* (community mental health), Heart-mind screen (Nepal)*, |
| Disability / Somatic [n = 5] | WHODAS 2.0, Somatic complaints questionnaire, Sheehan Disability Scale, Recovery Assessment Scale, The tension scale*, |
| Community mental health/ quality of life/ recovery [n = 5] | Community Attitudes to Mental Illness, Psychological Outcome Profiles (PSYCLOPS), WHOQOL-BREF, Recovery Star, Engagement index* |
| Mental health service use [n = 3] | Vermont Mental Health Consumer survey, Canadian Health Care Evaluation Project questionnaire, Staff Attitude to Coercion Scale |
| Resilience [n = 5] | Resilience scale (RS-14), Resilience Scale for Adults (RSA), General Self-Efficacy scale (GSE), Rosenberg self-esteem scale, CSRI Empowerment scale |
| Social factors [social support, caregiver burden and intimate partner violence] [n = 9] | Burden Assessment Scale, Family Interview Schedule (FIS), WHO questionnaire on partner violence, Multi-dimensional Scale of Perceived Social Support (MSPSS), Manchester Short assessment of Quality of Life, Social outcome index, Medical outcomes social support study, Zarit Burden Interview*, Mason empowerment scale* |

NB Scales developed in South Asia denoted with an asterisk

measures, and types of groups, as well as the diverse nature of the study contexts. Table 3 summarises quantitative measures used in the papers reviewed.

The key outcomes and enabling contextual factors of psychosocial groups are summarised in Table 4.

The primary outcomes of psychosocial support groups, as reported across both qualitative and quantitative studies, can be grouped into intrapersonal, interpersonal and community themes:

**Intrapersonal.** Nearly all studies that used quantitative measures reported significantly improved mental health and wellbeing for group members using pre-post comparisons and when measured against control groups. Some studies observed no significant improvements in mental health and wellbeing which was attributed to under-powered sample size [58], use of non-validated measures [56] and improvements that lasted only a short time [65, 69] or use of minimally trained peer group facilitators [63]. Due to the large variation in outcome measures, a meta-analysis is not feasible.

**Interpersonal.** Studies reported improved peer interaction, which increased rehearsal of mental health skills [61, 62] and increased perceived social support from other group members [18, 52, 62, 63, 68, 72], proposed to be a key enabling mechanisms. Another outcome described was improved financial status and income generating in households [62, 72, 75, 78–80].

**Community.** Studies described increased trust and social relationships in groups which led to increased social confidence, improved communication skills, and improved social network of people to call on during a crisis [18, 22, 52, 62, 63, 68, 75, 77–80, 87]. They also described other pragmatic benefits like sharing of knowledge on where to access resources or direct sharing of resources between group members [52, 62, 68, 72, 75, 80].

Both reference panels endorsed these findings. The Practitioner panel underlined that groups can spread limited mental health resources further, thus being more cost-effective than individual therapy, while the EBE panel additionally proposed that groups can increase social inclusion and supportive relationships/ friendships and are more fun to attend than individual psychotherapeutic sessions. These findings were supported by quantitative measures, which found that groups improved attitudes and awareness of mental health problems among the community.

**Table 4. Key outcomes and enabling contextual factors of psychosocial groups in South Asia.**

| Intervention | Summary of intervention studies (excluded abstracts and protocols) | Outcomes observed | Enabling factors in group intervention | Enabling contexts | Challenges (contexts and intervention) | Proposed mechanisms (author–posited) supporting outcomes |
|---|---|---|---|---|---|---|
| **Widows in North East India [22, 67]** | Support groups were formed among widows of intravenous drug users [IDUs]. Resulted in significant improvements in mental health, quality of life and somatic symptom metrics. | • Participatory Action Groups lead to improved feelings of social inclusion, physical health, economic capacity and sense of purpose and decreased discrimination<br>• Significant improvements in mental health symptoms and reduced somatic symptoms<br>• Increase in all quality-of-life measures, some significant but not all.<br>• Raised awareness | • Opportunity for widows to come together around the theme of mental health promotion provided reason to gather<br>• Awareness of mental health as an important health issue for all increased.<br>• Generous allowance helped recruitment and retention | • Not significant results in Nagaland may be explained by higher Quality of Life scores at baseline and smaller sample size. | • Retention: Overall retention rate of 80% is high considering the stressfulness and unpredictable nature of widows' lives<br>• Diversity in groups presented challenges in data collection and intervention delivery: Homogeneity of future groups recommended.<br>• Difficulty gathering sensitive information such as engagement in HIV risk behaviours | • Fun activities such as games and singing increased enjoyment and attendance<br>• Group helped overcome social isolation and provided a sense of belonging.<br>• Benefits of building relationships with other women in the same situation<br>• Improved relationships with parents, children and in laws.<br>• Better able to manage and respond to enacted stigma and discrimination.<br>• Group members collectively able to reach out and advocate for others. |
| **Service users in Gujarat, India [19]** | The WHO QualityRights programme implemented rights-based practice in specific mental health services. A component of this was self-help groups for service users and carers. Overall the programme had a positive effect on the quality of services. | • Significant improvements in service quality and adherence to QualityRights standards in services which received the intervention compared to those which did not.<br>• Significant improvements in staff attitudes towards service users and less use of coercion [intervention]. | • Engagement from staff at all levels, including involvement of senior staff in design and delivery. | Mental Healthcare Act 2017 in India was adopted during the study which is based upon and covers all the rights promoted in the QualityRights programme. | Staff turnover and changing cohorts of service users presented challenges in collecting and interpreting data. | Not reported. |
| **People with mental distress in Uttarakhand, India [62]** | Intervention included individual component and invitation to participate in psychosocial support groups– 60% of participants joined these | • Significant improvement in depressive symptoms, disability, recovery and community engagement for people with common and severe mental health problems as well as epilepsy | • Responsive to group priorities e.g. training in access to Government entitlements<br>• Building knowledge through dialogue increased knowledge coproduction and challenged biomedical hierarchies<br>• Groups discussed local frameworks to increase acceptability of knowledge | • Group meetings held in homes which enabled women to attend<br>• Increased freedom of movement challenge to gender hierarchies<br>• Link to micro-credit and savings increased participation by women with limited freedom of movement | • Patriarchy limited freedom of movement and opportunity to attend groups for some women<br>• Mixed group membership including caregivers and people with mental distress maybe reduced participation for some | • Value of rehearsal of social skills in group to increase mental health and to increase social inclusion<br>• Possible improved financial security through micro-credit and savings may have improved mental health<br>• Components addressing social determinants of health addressed what mattered and increased relevance for participants |
| **Participatory groups with mothers across South Asia [52, 68, 69]** | Tripathy et al., 2010 [68]: Groups of mothers of newborns were formed with the aim of improving birth outcomes and maternal mental health. Compared to existing women's groups or baseline, there was a significant reduction in newborns deaths and a marked reduction in symptoms of depression in mothers later in the study. | • Significant reduction in neonatal mortality rates [45%] in intervention compared to control groups. No differences in help-seeking behaviour.<br>• No detectable difference in maternal depression scores for severe depression, however, in year 3, a 57% reduction in moderate depression among intervention mothers. | • Simple process for randomisation helped build trust through transparency.<br>• Health committee members knowledgeable about government health system, assisted with formation of village health committees | Differences in maternal education, tribal membership and assets between the intervention and control populations. | | As well as improving social support, the shared experience of groups may have helped with problem-solving skills and practical solutions for therapeutic benefits. |
| | Clarke et al., 2014 [69]: Groups of mothers of newborns were formed in order to develop strategies to improve health for themselves and their child, including perinatal common mental health problems. | No significant difference in postpartum psychological distress between intervention and control groups. | • Facilitator respected source of information, participants share knowledge in community, increased acceptability of recommended neonatal health practices, good social support among group members.<br>• In rural Bangladesh women are secluded which may have prevented engagement with women in a generally empowering way and reduced psychological distress. | • Social barriers to engaging with other women ie. Limited freedom of movement<br>• Rural Bangladesh exposed to environmental stressors like flooding<br>• Participation of young, newly married women in the groups was difficult in some areas.<br>• Local people recruiting were vital in maintaining effective relationships with the local community | • Participation lower among mothers living in poorest households, primigravid, younger, secondary education or above. [2014].<br>• Staff turnover high<br>• Family constraints on freedom of movement<br>• Experience of delivery of the intervention on a smaller scale enabled PCP to meet the majority of requirements for successful scale-up | • Possible explanation for null result in this setting is that mental health benefits of • • • •Women's groups were not fully realised in Bangladesh.<br>• Neonatal mortality may be less important as predictor of postpartum psychological distress in Bangladesh compared to eastern India. |
| | Morrison et al., 2019 [52]: A qualitative study with women who attended groups to support mothers of newborns across all studies in this cluster to further explore their mechanisms and impact. | Women's group increased knowledge about maternal and newborn health problems; provided social support; and increased confidence to act which all stimulated behaviour change. | • Facilitators were a respected source of information<br>• Participants share knowledge in community<br>• Women promoted and sold safe delivery kits to increase availability and improve preparedness among poor women | . •Economic support increased confidence of women<br>•Home-care behaviours that were low-cost increased engagement of poorer families<br>• Good social support among group members. | •Communities in rural Bangladesh strongly patriarchal and challenges include gender-based victimization, including domestic violence and marital problems<br>•Restrictions preventing women from going outside the home were barriers to accessing information that would be beneficial for their health and the health of baby | •The process of learning and action increased confidence of women so they could negotiate in families for behaviour change.<br>•Women learned from each other reinforcing ideas and behaviours, making it easier for families who were poorer or less educated to learn.<br>•Acceptability of healthy behaviour increased due to conversations and role-modelling across the community<br>•Community respect increased so groups recognised as a community resource. |

*(Continued)*

**Table 4.** (Continued)

| Intervention | Summary of intervention studies (excluded abstracts and protocols) | Outcomes observed | Enabling factors in group intervention | Enabling contexts | Challenges (contexts and intervention) | Proposed mechanisms (author–posited) supporting outcomes |
|---|---|---|---|---|---|---|
| **Women in self-help groups in India** [72] | A mental health intervention, involving group counselling and stress management, was integrated into existing women's micro-credit self-help groups. Qualitative results suggested improvements in sleep quality and social support, and a decrease in psychological distress. | • The majority of participants [N = 18/21, 86%] reported that the quality of their sleep had improved with regular practice of the relaxation exercise • Women described feeling unburdened and "lighter" with greater feelings of trust and kinship. • More women in the mental health intervention group had started their own small business. | • Women reached out across social boundaries. • The experience of the senior members of the group and knowledge of younger ones generated a good mix of coping strategies • Group members were able to self-sustain groups later on. | N/A | • Women did not open up during focus group discussions but tended to agree with the first speaker. • Difficulty finding a meeting space which was large and private. • Some women stated that they were not able to practice the relaxation at home due to time and space constraints • Lack of literacy skills made using quantitative metrics difficult. | • More personal interaction through the mental health intervention, and associated increases in feelings of trust and reciprocity, may have increased social capital • Capacity building using SHG model strengthened women's ability to generate income. • Addition of mental health intervention to SHG microcredit activity led to greater interpersonal trust and a stronger social support network. • Contributing to family income resulted in women commanding more attention and respect within the household and participating in decision making processes. |
| **Women surviving tsunami in India** [53–55, 57] | Groups of women survivors of the 2004 tsunami disaster in India were formed with the aim of providing psychosocial support across a 3-month period. Compared to a control group, there were significant improvements in symptoms of emotional distress in the mental health intervention group. | • Significant improvement in emotional distress related to the tsunami event for those in the intervention groups. • Intervention groups scored with significantly less distress than control groups at end of trial. | • Some trainees were also survivors so had greater empathy, and as locals could speak the same language and share cultural traditions. • Train-the-trainer model very effective for training large numbers of community-level workers. | Not reported | • Logistical delays due to working in disaster response context. • Some trainees delivering intervention are also survivors so loss of objectivity in providing services. • Some dropouts after initial recruitment due to 'cultural restrictions', e.g. husbands or extended family not permitting women to be part of groups. | Not reported |
| **Women in North India** [18, 42, 73, 74] | Qualitative case studies of an existing community mental health project involving psychosocial groups for local women with common mental health problems. Effects include reduced stigma as well as improvements in mental health and mental health understanding, both within and beyond the groups. | • Groups increased social and emotional support for participants. • Created a safe social space for women where they could 'speak freely' • Improvements in mental health for group members. • Improved knowledge about mental health and decreased stigma, including for caregivers. • Participants modelled respectful communication. | • Engaging families was a more effective strategy to ensure women's participation • Participation easier for employed and more educated women • 'Corner conversations' and role plays with peer-to-peer knowledge sharing seemed better than large group meetings for engagement and emergence of new knowledge | • Groups with strong peer relationships led to increased capacity for collective action • Pluralist approach where new knowledge added to existing explanatory framework led to higher levels of acceptance | • Principal barrier to participation in groups was women being unable to leave home and stigma from other community members. • Families prioritise income generating activities. | • Psychosocial groups provided new social networks which could act as social support, especially for members who were previously very isolated. • Women build strong, trusting relationships with one another which enabled them to share personal information and issues. • Members felt more knowledge of mental health led to greater ability to cope with problems individually or as a group. • Personal determination of group participants to leave house and distance to support group are important, especially when acting against conflicting gender norms. |
| **Women with depression in Bangladesh** [75] | Groups with 12 women with depression were formed and met fortnightly over 6 months with trained peer facilitators following an intervention for managing depression and increasing financial literacy; the intervention included formation of 'bandhobi' pairs to promote friendship. | • significant improvement for intervention group in PHQ9 [depression measure] score and improvement in multiple other social and mental health measures including a cultural measure of distress, self-esteem social support and hope/future orientation and financial decision-making and coercion [measures for each described in paper]. • Mixed results in Empowerment scale | • High attendance and retention linked to economic strengthening with financial reward for savings; • Link with economic strengthening may create therapeutic synergy, enhancing autonomy | • significant improvement in PHQ9 [depression measure] score | Economic strengthening proposed as able to empower and improve social status for women | • The bandhobi pairs components let to formation of enduring peer friendships for half of women in intervention—increased engagement and improved mental wellbeing. • may have improved mental health through mediators such as hope, self-esteem and social support |
| **Adults in Nepal** [60] | People diagnosed with epilepsy or psychotic disorders were either given a 'comprehensive' package of care, which included psychosocial interventions such as counselling and peer support, or medication only. There were significant improvements across all sub-groups, and reduction in epilepsy symptoms was the only significant benefit of a 'comprehensive' package of care. | • Patients diagnosed with a psychotic disorder: significant improvement in psychotic and depressive symptoms, function and social behaviour in both intervention and control groups. • Patients diagnosed with epilepsy: significant improvements in depressive symptoms, functional impairment and familial burden in both intervention and control groups. | Not reported. | Not reported. | • Limited sample size inhibiting ability to make definite conclusions | Not reported. |

(Continued)

**Table 4.** (Continued)

| Intervention | Summary of intervention studies (excluded abstracts and protocols) | Outcomes observed | Enabling factors in group intervention | Enabling contexts | Challenges (contexts and intervention) | Proposed mechanisms (author–posited) supporting outcomes |
|---|---|---|---|---|---|---|
| Problem management Plus [PM+] [58, 61, 92]. | A community-based group version of the WHO Problem Management + intervention was trialled to test feasibility. Participants were women with identified mental health problems. Results suggested robust acceptance and positive effects on mental health [92] | • Improvements in depression and anxiety symptoms, functionality, psychological outcomes, PTSD symptoms and PCQ-9 generalised distress for the intervention group. Study not sufficiently powered to detect significance. | • Participants learnt useful skills to manage their problems. • LHWs were respected and trusted in the community and linked Group PM+ facilitators/delivery agents and women in the community, overcoming a significant access barrier. | • Groups gave participants a safe space to share their feelings as they problem solved together. • Local facilitators more likely to understand people's problems and be culturally sensitive than therapists from outside community. | • Confidentiality issues. • Monetary expectations. • Long duration of sessions • Lay counsellors initially not competent but addressed through additional targeted training. | • Group format not only draws on same strategies as individual version of PM+, but also has the benefits of peer interaction and potential for group members to be therapeutic agents to each other. |
| | A community-based group version of the Problem Management + intervention, initially trialled in rural Pakistan [Khan et al., 2019] was then implemented in more conflict-afflicted areas of the same setting. Participants were women with identified mental health problems. Improvements reported in symptoms of anxiety and depression, PTSD and functionality. [61] | • Measured 3 months after intervention, there were significant improvements in anxiety and depression symptoms, functionality and problems for which people sought help in intervention groups compared to control groups. PTSD symptoms were significantly improved at 1 week post-intervention but not 3 months. • Non-significant improvements were observed in perceived social support. | • Cascade model of training between supervisors and facilitators. • Supervision taking place virtually. | Effects of conflict more diffused and lower baseline scores related to PTSD symptoms. Ongoing problems of living, rather than war, has more of an effect on these women's lives. | | Not reported. |
| | A community-based group implementation of PM+ which was adapted for Nepal in the post-earthquake context. Participants were adults with common mental health problems in Sindhuli district, Nepal. [58]. | • Demonstrated high level of feasibility, participation and acceptability. Though under-powered for effectiveness, mean improvement of depression greater in intervention than control group with improvements in other 4 measures also. | • Group format reported to reduce sense of isolation and increase awareness of CMD • Helpful to gender-match facilitators and data collectors to participant gender | • Recruitment assisted by local name of programme not stigmatising idioms 'Open/non-stressed mind' • Expectation of financial remuneration to participate maybe increased dropout | • Groups with men had initial hesitancy sharing personal problems, busy work schedule and difficulty meeting due to income generation outside of communities. • Lack of psychiatric treatment seen as negative by participants | • Use of local idioms helped normalise mental distress experiences making participants open to PM+ content • Widespread women's group in Nepal normalised participation for women and they found it fine and mood lifting |
| Tsunami survivors in the Andaman Islands [56] | A group-based yoga program was used as an intervention for survivors from the Andaman Islands of the December 2004 tsunami in India. There were significant improvements in self-reported indicators of distress and physiological measures of heart and breathing rate. | • Significant improvements in indicators of distress. • Significant improvements in breathing rate, and significant improvements in heart rate for participants who were mainland settlers. | Not reported | Not reported. | Not reported | • Yoga practice has been found to have beneficial effects on sleep quality which could explain other beneficial psychological effects. • No correlation was found between measures of breath rate and indicators of psychological distress. |
| Social support among Pakistani women [77] | Women with no pre-existing mental health diagnosis partook in either a 6-week structured social support group [intervention] or attended a single mental health awareness session [control]. Compared to the control group, women in the intervention group showed significant improvements in resilience but no differences in quality of life. | • Measured 6 weeks post-intervention. • Members in the social support intervention reported significantly greater improvements in resilience, through RS-14 and a sub-scale of RSA, compared to the control group. • Improvements in WHOQOL-BREF scores for both groups, but no significant differences. | • Women experienced a safe and trustworthy environment and began to openly share their experiences after a few sessions. • Peer leader emerged in every group who was emotionally strong and shared experiences and advice. • Women discussed ways to achieve better family life and better future. | • Context of gender relations restricting women from talk openly about their needs and goals in life. • RSA scale used as it has more social and family domains of resilience which were deemed to be more gender sensitive. | No further follow up on results meant it was hard to determine whether positive effects would be sustained, especially after the one-off control group session. | • Women in the group experienced a safe and trustworthy environment, enabling them to reflect and learn more openly. Supported by reminders about confidentiality. • Suggest women take a 'tend-and-befriend' approach to dealing with life's stressors, which builds social ties. • Women seemed to prioritise the wellbeing of their families which may have meant that support offered by the group resulted in them feeling more capable in making beneficial decisions for their families. |
| Economic skills building among Pakistani women [78–80] | Women were randomly assigned into one of three community-based groups: counselling, economic skill building [ESB] and a control. Each group received an 8-week intervention programme. Results suggested that women in the ESB group showed significantly higher self-efficacy, but non-significant improvements in employment rates, depression and partner violence. | • Non-significant lower depression scores in ESB group [20.1] compared to counselling [24.7] and control [27.6] groups. • Significantly higher self-efficacy in ESB group compared to counselling and control groups. Non-significant improvements in employment rates, depression and partner violence in ESB group compared to counselling and control groups. | • Community-based participatory approach in developing the ESB intervention, as well as the location and time of interventions. | • Starting with a focus on literacy program supported development of trust with male family heads • Incorporating mental health components into literacy program was less stigmatising & led to greater participation • Group mental health focus on problem solving addressed felt need | •Stigma linked to mental health program •Implementation in homes not possible due to privacy concerns and challenges to discuss gender-based violence •Local women as facilitators were limited in their role by patriarchal husbands–additional support for some Lady Health workers increased their effectiveness | •Implementation linked to literacy groups legitimised attendance and minimised stigma leading to greater engagement with mental health components • Trust within groups led to problem solving e.g. employment and economic self-sufficiency problem solving in groups •Local women as Lady Health workers meant they used language, and examples that were relevant for participants, increasing engagement and participation. |
| Adults with depressive symptoms in Pakistan [64] | Set in urban Karachi, 30 people w depression/ anxiety and their caregivers participated in monthly multi-family group meetings facilitated by lay community member for 6 months, assessed by mixed methods evaluation | • Significant improvement of subjective quality of life, self-reported anxiety and depression scores, social outcome index but no change in family burden index. | • Group provided a platform for shared learning, helped participants feel better at self-regulation and provided a sense of belonging [qual data] | • Facilitation by trained community members may have given a sense of less hierarchy and increased ease in communication | • No improvement in caregiver burden may be due to dialogue which underlined carer vs patient roles • Many participants described this as a class with facilitator as 'expert' | • Participants described improved confidence communicating and social skills • Group activated positive behaviours and social networks within families and their wider community |

(*Continued*)

**Table 4.** (Continued)

| Intervention | Summary of intervention studies (excluded abstracts and protocols) | Outcomes observed | Enabling factors in group intervention | Enabling contexts | Challenges (contexts and intervention) | Proposed mechanisms (author–posited) supporting outcomes |
|---|---|---|---|---|---|---|
| **Thinking Healthy programme** [63, 83–89] | The Thinking Healthy Programme [THP] is a cognitive therapy-based intervention addressing perinatal depression in low resource settings. It has individual as well as group sessions delivered during pre- and post-natal care. It improves perinatal depression, as well as social functioning and overall disability [61]. It has since been adapted for delivery by peers [THPP] but was not found to have significantly beneficial effects over controls. | • The studies in the group showed a range of findings. The Nusrat study described significant improvements in mental health; [86] while the Sikander 2019 paper found THPP had no effect on symptoms of depression or remission but significantly benefitted disability and severity metrics [63]. Maselko study found reduced symptom severity and remission in both arms perhaps masking treatment effects [89] <br>• Implementation strength ranged: 77–96% [88] <br>• There was strong relationship between disability score and implementation strength but no association with PHQ9 [depression score] outcomes [88] <br>• A trial examining effects of Sikander trial 3 years post intervention found no difference in PHQ[Depression] score between intervention and enhanced usual care group [89] | • Arranging sessions at convenience of mothers. <br>• Intervention adapted for peer delivery, including simplified focus on behavioural activation and information by narrative or vignette. <br>• Field supervision enhanced credibility of peer volunteers in community. <br>• For peer volunteers: helping others, learning new skills, feeling appreciated elevated community status. <br>• Desirable peer volunteer characteristics [local, ability to form trustworthy and empathetic relationship with the mother and family]; <br>• Important to link with local primary healthcare system. | • Close-knit communities. <br>• Despite lack of financial renumeration, retention of peer volunteers was high. Perhaps through a form of social compensation, providing opportunities for the women <br>• Groups preferred facilitators who were middle-aged, educated with similar experiences to participants and good communication skills and good character | • Cultural restriction on women's mobility, eg.chilla ritual [40-day confinement period after childbirth]. <br>• Mother's engagement: mothers unwilling to disclose personal information when peer volunteers live in the same village <br>• Motivation of peer volunteers: lack of mother's engagement and resistance from mother's family; ambiguity of peer volunteer role; <br>• Family beliefs that mothers should seek advice from female elders. | • Local peer volunteers preferred because of their understanding of cultural norms however outcomes were greater when Thinking healthy was delivered by community health workers [higher training and experience than peers] <br>• Linking with primary healthcare centres •Enhanced credibility of peer volunteer role, providing greater sense of identity and ownership of the programme. <br>• Stigma addressed by framing intervention as targeting 'health' rather than depression. <br>•Shared life experiences of peer facilitators identified to increase group engagement <br>• Low frequency and low attendance of group meetings [twice a month] may have reduced impact of intervention |

## Mechanisms linked to outcomes

Some mechanisms, such as developing trust could be understood as both an intermediate outcome, as well as a mechanism. We identified five key mechanisms by which groups can improve mental health:

The underlying base mechanism that seems to trigger the subsequent four mechanisms is a sense of **being part of trusted relationships** with other group members. In Southern India, participation in self-help groups was identified as leading to increased interpersonal trust, which the study posited led to greater cooperation, increased social belonging and mental well-being [72]. Trust was described as improving the quality of conversations and relationship in groups, leading to more vulnerable sharing and collective problem solving [18, 72]. Another study also proposed these trusting relationships enabled group members to act as therapeutic agents to each other [65]. Data conversely supporting this was a short-duration group intervention in Bangladesh which failed to improve mental health measures among postnatally depressed women, whose authors proposed that the limited time in groups meant less building of mutual trust may have contributed to the lack of successful outcome [69].

A second mechanism that supported outcomes was that forming relationships which is facilitated through the group platforms triggered **a sense of inclusion and support**, which then meant they could access both physical and social resources and thus felt able to participate [52, 72]. For example, in a post-disaster situation, formation of groups helped with restoring social relationships and accessing resources [55, 56]. Similarly, Rao suggested that reciprocity and shared activity of a micro-credit and savings group accelerated trust between group members [72].

A third key mechanism **feeling able to manage mental distress** due to improved psychosocial skills and knowledge [18, 19, 22, 52, 72, 77]. Authors proposed that groups offered a safe space unburden and share problems [52, 62, 68, 72, 75, 80]., and also to learn skills in problem solving, rehearse communication and increase mental health skills. The safe learning space offered by the group environment led to greater participation and sharing of experience and therefore exposure to a broader range of coping strategies and problem solving from group members [18, 22, 52, 62–64, 68, 72, 75, 77, 80].

A fourth mechanism was that a group membership triggered **a sense of belonging.** Authors noted this sense of belonging and trust took time to develop meaning that relationships grew

stronger with time [18, 22, 52, 61, 63, 72, 77]. This sense of belonging triggered behaviour changes, for example, widows in North East India described that they were able to identify and control feelings of anger because of their sense of safety in the group [22].

A final mechanism was a **sense of collective strength particularly among women** [18, 72], which supported women to resist gender norms and to act autonomously and also take actions for advocacy for their own well-being [18]. This was described as then contributing to improved mental health [18, 22, 52, 63, 67].

The Practitioner panel underlined the value of increased mental health knowledge and skills to lead to outcomes while the EBE panel particularly underlined the value of increased social connection and peer friendships which was linked to a sense of trust and belonging. They also noted that homogeneous groups were better able to provide social support and friendship, which could then trigger improved mental health outcomes.

## Our program theory builds on the above findings and is summarised here narratively

In a context where people with psychosocial disability are socially excluded and have limited autonomy, psychosocial support groups can provide a safe space that facilitates a sense of mutual trust and the opportunity to participate in trusted relationship with peers. These relationships trigger a sense of inclusion and mutual social support, feeling able to manage one's mental distress, belonging and collective strength. This triggers group members to take collective action that addresses structural and intermediary determinants of health and leads to outcomes of improved mental health and greater social participation and inclusion. We note that the distinction between outcomes and mechanisms and lines of causation are blurred and often bi-directional. A simplistic schematic summary of the context, enablers, mechanisms and outcomes is presented in Fig 2.

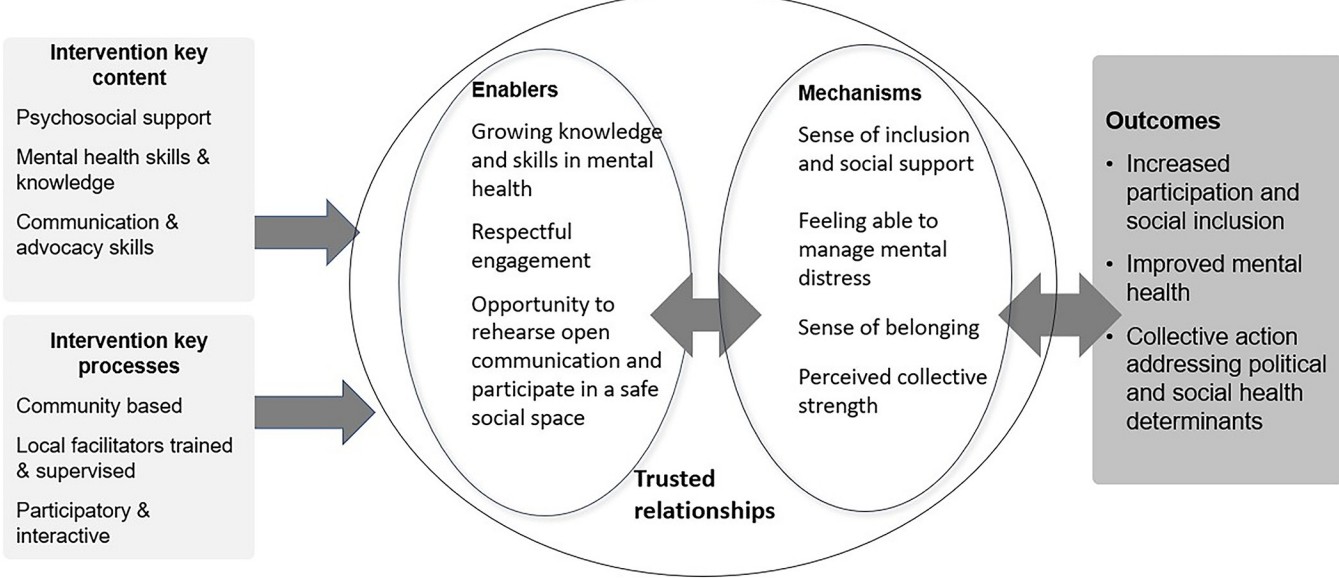

**Fig 2. Schematic representation of the context, mechanisms and outcomes linked to participation in psychosocial groups in South Asia.**

## Discussion

### The benefits of psychosocial groups

This review aimed to use a realist lens to investigate the use of psycho-social groups as mental health interventions in South Asia, looking at how groups were setup [demographics, facilitators, interventions] as well as measures, outcomes and proposed mechanisms.

There was evidence that group interventions can have multiple positive effects for individual participants. This included improvements in mental health and wellbeing, social inclusion, self-efficacy and, depending on the nature of the group, financial position. Many of these benefits centred on the relationships in the groups, which enabled sharing of coping strategies and rehearsal of social skills, as well as access to an increased social network and, by extension, social support which could improve social capital as well as provide a sense of belonging [18, 52]. Trust seemed to be integral to realising these benefits, both in terms of trusted relationships between group members and trust in the group setup itself, for example that the group was facilitated in a way which was confidential and sensitive to local dynamics, and trust seems likely to be a core requirement for all group mental health interventions in this context. This review highlighted that local facilitators may have an advantage in understanding these contextual dynamics and ensuring group content is relevant [52]. The presence of government incentivised community health workers across South Asia (for example, Lady Health workers in Pakistan, Auxiliary Nurse Midwives and Accredited Social Health Activists in India) offers a vast opportunity for facilitation of group interventions which has barely been explored [93].

Findings point to a connection between these benefits and duration of group participation. However, some of the longer interventions suggest benefits plateau after six to eight months [52, 68, 69]. It may be that peer friendships and collective action are established and continue informally after a few months regardless of formal meeting structures. Further research needs to examine the duration of groups and how and whether new peer relationships are sustained, and how.

Beyond individual effects, this review identified that group participation seemed to build an ability to act collectively. This has also been found in a meta-analysis examining groups among pregnant women in South Asia [26]. Acting collectively is a key component of mental health competent communities [42, 94–97], facilitating community engagement, advocacy, social inclusion and improved economic status [98, 99]. Thus, in connecting the individual to the social and in their ability to improve capacity for community action, psycho-social groups can influence the structural factors contributing to mental ill-health, both responding to mental health needs and reducing future need [100].

These potential benefits are increased by the fact that groups seem to be a socially acceptable form of intervention in various community settings of South Asia, and some evidence suggested they were a preferred form of psychosocial support, for example compared to individual counselling for women in North India [18]. Additionally, group interventions may be more accessible for certain marginalised groups, such as widows of intravenous drug users in of our reviewed studies [22] and women in low income communities [18], who otherwise may face barriers due to stigmatisation. However, our literature search did not identify interventions targeting minority groups such as LGBTQ+, Dalit, or disabled individuals and further research should examine barriers to participation for such groups. Group intervention also allow a larger number of people to access interventions at one time meaning they are likely to be more cost effective than individual therapies, especially when facilitated by lay community members [101], however further research needs to examine the cost-effectiveness of group vs individual interventions.

## Gender and psychosocial groups

The majority of studies in this review implemented psychosocial groups among women, although this may have been skewed by our exclusion of studies linked to alcohol and substance abuse. Women are systematically disadvantaged across South Asia, for example India is ranked 140[th] and Pakistan 153[rd] out of 156 countries in the gender parity index of the Global Gender gap report in 2021 [102]. This structural gender inequality is a key contextual mental health determinant, negatively affecting women's wellbeing and access to care [103, 104]. Our review identified that for women, psychosocial groups are accessible, acceptable and feasible to participate in, with additional functions such as microfinance making participation in groups even more legitimate and practical, linked to greater financial stability and prestige associated with providing income in the household [52].

However, almost no studies included men, despite evidence that men benefit from participation in psychosocial groups in high income settings [105]. Gender norms that confer status on men also carry risks for poor mental health. In South Asia, men are more likely than women to engage in interpersonal violence, harmful use of alcohol and other drugs, and suicide [106]. Men are also less likely to acknowledge vulnerability or anxiety and thus to seek help with health professional or peers [107, 108]. Gender plays a critical role in mental health for both women and men, and this review identifies that addressing gender relations through psychosocial support groups as well as through other interventions is central to promote mental health in the South Asian region.

## Context-specificity, heterogeneity and methodology

South Asia presents diverse settings. This study was motivated by a question posed by a local organisation in North India. A research question with a local pose and gaze means it is more likely to be relevant and to lead to practical applications [43]. The dominance of Euro-America and biomedicine in global mental health are reflected in the fact that just 8 of 46 measures used by researchers in this review had been developed in South Asia. Further research may benefit from comparing results from locally developed measures with those from measures developed elsewhere in order to ascertain whether measures capture the priorities of communities within the SAARC context.

## Implications

This review has important policy and research implications. It demonstrates psychosocial groups as highly cost-effective [26] and improving health and social outcomes within existing health system structures [101, 109]. Rolled out at scale, psychosocial groups can provide policy makers with a core strategy to increase the reach and effectiveness of current mental health services. For example, the mechanism of increased sense of ability to manage distress suggests the need to prioritize active learning in a way that people can apply in their own local contexts. Thus, psychosocial groups can address contextual issues that affect mental health and which cannot be effectively resolved in one-on-one therapy alone.

The value of psychosocial groups to our knowledge has not been codified in national mental health policy in any South Asian nation. India's National Mental Health Programme through its District Mental Health Programme [DMHP] includes a [psychiatric] social worker as part of the team, although this is constrained by the biomedical orientation of the DMHP [30, 110]. Similarly, in Bangladesh, Nepal and Sri Lanka, a key priority is building capacity in mental health skills and knowledge for community health workers [111–113]. While micro-credit and savings or "Self-help groups" as well as farmers groups, operate widely in South Asia and could

be tapped as a community resource, they would require resources including training and orientation from District Mental Health Programme mental health personnel.

Further research is needed to further examine the contribution of psychosocial groups in mental health globally, including among men. Randomised controlled trials typically lacks information about contexts and mechanisms which are often the most important determinants of local mental health outcomes [9, 43, 94, 114, 115]. Qualitative companion papers can provide detail on what works, for whom, under what circumstances (eg. In this review, Morrison et al., 2019 [52]). Global mental health and research that evaluates complex interventions requires methodologies that engage with the critical knowledge held by communities, and these are typically qualitative [43, 116].

## Strengths and limitations of the methods

There are several methodological limitations: First, it was difficult to describe psychosocial groups appropriately in the search. It is therefore possible that we excluded studies where psychosocial aspects of intervention groups had significant mental health benefits but were not described as such, for example, group interventions that address alcohol and/or substance abuse. Second, the realist lens we hoped to use in this review was constrained by limited information in our included papers which rarely described contexts and mechanisms [33, 117]. Without this information, we were unable to fully analyze the factors that contributed to the success of psychosocial interventions. Third, while we reviewed literature on SAARC countries, our stakeholder engagement was limited to India due to a limited budget which precluded travel to conduct reference panels in other South Asian countries. Contributions from researchers in other SAARC contexts may offer further insight into the commonalities and differences that facilitate the benefits of psychosocial interventions within local communities. Fourth, we did not specifically examine or identify from the selected studies, how psychosocial group interventions work for minorities groups such as LGBTQI or oppressed caste groups. Further research will be necessary in order to determine whether such groups receive equal benefits from similar mental health interventions. Fifth, recognising that the majority of psychometric measures used in this study were developed outside of South Asia, the measures may be less accurate and relevant for this setting compared to the place where they were first developed.

Strengths of this study include our effort to include grey literature (and the associated learnings from non-academic practitioners), and triangulation of the findings with reference panels [practitioners and EBE]. These efforts allowed us to include perspectives that are not always represented within biomedical research, but which can help to shed light on the effects of various intervention methods within a community. Additionally, like Daudt et al. [118], we started with a broad question and we had to narrow down the scope of the work, clarify relevant concepts, and constantly review our inclusion criteria. This flexibility ensured that the focus of our research could be as specific as possible, as central concepts were repeatedly reviewed to ensure relevance for our topic. A further strength was the transdisciplinary team which include people specialised in public health, social work and psychiatry backgrounds, social work and health informatics backgrounds. These diverse perspectives offered insight into the different considerations surrounding mental health interventions, including their interplay with cultural context and practical issues such as the cost of implementation.

## Conclusions

This is the first review to our knowledge to examine the contribution of psycho-social groups as a mental health intervention in low- and middle-income countries. We examined 15

interventions in South Asia and found they were almost universally acceptable, relevant, and effective in improving mental health. Psycho-social groups work across a wide range of contexts in South Asia and further, can potentially address social and structural causes of mental ill-health. The group format appears to support several interconnected mechanisms that build on trusting relationships between group members, and integrate increased individual self-efficacy with collective action can improve mental health and social participation.

Despite the evidence of group interventions as effective and scalable, no national mental health policies in South Asia identify psychosocial groups as a platform to provide care to our knowledge. Psychosocial groups as a means to promote and improve mental health merit further research and policy attention and have significant potential to contribute to contextually relevant mental health care in South Asia.

## Supporting information

**S1 File. Summary of group participants.**
(DOCX)

**S2 File. PRISMA checklist for scoping reviews.**
(DOCX)

**S3 File. Sample of MEDLINE search.**
(DOCX)

## Acknowledgments

Appreciation for the time and critical reflections shared by the Experts by Experience group members from Kanwali road and Brahmanwallah communities in Dehradun, Uttarakhand and also appreciation to the Technical Experts in Community mental health in India group members who participated in the meeting held in New Delhi on 2nd May 2019 which included board certified psychiatrists, psychologists, psychiatric social workers and leaders of non-profit community mental health organisations: Alok Sarin, Anish Cherian, Bhargavi Davar, Jagadisha Thirthalli, Prasad, Rajeshwari, Triptish Bhatia, Madhu Juneja, Pallab Maulik, Satabdi Chakrabarty

We are also thankful to Pooja Bhatt, Kakul Sairam, Jeet Bahadur, Laxman Balan, Samson Rana and Atul Goodwin Singh for support in this project as well as ongoing administrative support from Herbertpur Christian Hospital. Marshall Dozier supported search strategy development, and Claudia Pagliari commented on an early version of the study protocol.

## Author Contributions

**Conceptualization:** Kaaren Mathias, Maria Wolters.

**Data curation:** Meghan Davis, Maria Wolters.

**Formal analysis:** Kaaren Mathias, Sumeet Jain, Robert Fraser, Meghan Davis, Rita Kimijima–Dennemeyer, Pooja Pillai, Smita N. Deshpande.

**Funding acquisition:** Kaaren Mathias, Maria Wolters.

**Investigation:** Kaaren Mathias.

**Methodology:** Kaaren Mathias.

**Project administration:** Kaaren Mathias.

**Resources:** Maria Wolters.

**Software:** Maria Wolters.

**Supervision:** Smita N. Deshpande.

**Validation:** Pooja Pillai.

**Writing – original draft:** Kaaren Mathias.

**Writing – review & editing:** Kaaren Mathias, Sumeet Jain, Robert Fraser, Rita Kimijima–Dennemeyer, Pooja Pillai, Smita N. Deshpande, Maria Wolters.

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
