## [Decision Letter · Decision Letter 0]

6 Jan 2023

PGPH-D-22-00857

FULL Addressing mental ill-health through psycho-social groups in South Asia – a scoping review using a realist lens

Dear Dr. Mathias,

Thank you for submitting your manuscript to PLOS Global Public Health. After careful consideration, we feel that it has merit but does not fully meet PLOS Global Public Health’s publication criteria as it currently stands. Therefore, we invite you to submit a revised version of the manuscript that addresses the points raised during the review process.

We kindly recommend:

Clarifying and correcting the review methodology by following the PRISMA guidelines for scoping review.Reconsidering or consistently applying the realistic approach.Explaining what kind of evidence was extracted from what type of research.Critically evaluate the Discussion section.Organizing references to- and clarifying the content of supplementary files.Paying particular attention to reviewers’ comments (below) and implementing amendments in detail.

We look forward to receiving your revised manuscript.

Kind regards,

Hanna Nalecz, Ph.D.

Academic Editor

Journal Requirements:

2. Please provide separate figure files in .tif or .eps format only and remove any figures embedded in your manuscript file. Please also ensure that all files are under our size limit of 10MB.

3. Tables should not be uploaded as individual files. Please remove these files and include the Tables in your manuscript file as editable, cell-based objects. For more information about how to format tables, see our guidelines: 

https://journals.plos.org/globalpublichealth/s/tables

4. We have noticed that you have uploaded Supporting Information files, but you have not included a list of legends. Please add a full list of legends for your Supporting Information files after the references list. 

Additional Editor Comments (if provided):

Reviewers' comments:

Reviewer's Responses to Questions

**Comments to the Author**

1. Does this manuscript meet PLOS Global Public Health’s publication criteria? Is the manuscript technically sound, and do the data support the conclusions? The manuscript must describe methodologically and ethically rigorous research with conclusions that are appropriately drawn based on the data presented.

Reviewer #1: Yes

Reviewer #2: Partly

2. Has the statistical analysis been performed appropriately and rigorously?

Reviewer #1: N/A

Reviewer #2: I don't know

3. Have the authors made all data underlying the findings in their manuscript fully available (please refer to the Data Availability Statement at the start of the manuscript PDF file)?

Reviewer #1: Yes

Reviewer #2: No

4. Is the manuscript presented in an intelligible fashion and written in standard English?

Reviewer #1: Yes

Reviewer #2: Yes

5. Review Comments to the Author

Reviewer #1: Thank you for the opportunity to review this paper. The paper aims to summarise what we know about interventions that target mentally ill adults at group-level, are delivered by community health workers and with a clearly defined psychosocial component.

The full title of the paper does not reflect the aforementioned summary well enough and could be ambiguous (…through(?) psycho-social (support?) groups or psycho-social group interventions(?)).

The link between initial search in 2019 and the inclusion of studies till February 2022 is confusing in the Abstract (clear in the paper).

Authors assert that they use the realist approach for at least part of the study. Among the 3 questions that they outline on p.4, q3 is the realist component. The question in part is: “what enables these mechanisms to trigger positive outcomes?” In the realist view, mechanisms are the ones that trigger action (outcomes) within agents (in specific contexts). The current phrasing needs to be adapted to make it consistent with the the understanding of mechanisms in realist inquiry. Or else this inconsistency ought to be explained/discussed.

Among the mechanisms identified, the first one could benefit from a critical discussion among authors. I believe the framing of the mechanism ought to communicate the tendency within people that triggers action in some contexts (and not in others). The first mechanism appears to be the direct result/outcome of the actual mechanism. Please review. Compliments on the framing of the other mechanisms which appear very useful in program design.

On p.13, reference is made to fig 1 which is supposed to be a middle range programme theory, but fig 1 is PRISMA on p.8. Authors possibly refer to fig 2.

There appears to be some conceptual confusion between middle-range theories & programme theories. There are instances when program theories can indeed be middle-range theories if they have sufficient degree of abstraction. However, what the authors attempt to do in figure 2 appears to be a comprehensive identification of intervention inputs/components, enablers, mechanisms and outcomes. This is NOT a middle-range theory nor can it be a good (initial/refined) programme theory. Programme theories are typically either explanations/schematics that describe what works, for whom, under what conditions and why. IN a sense they do leverage one or more outcome configurations and are an abstraction of one or more CMOs. But an aggregation of different aspects of diverse interventions found in literature into one schematic identified as a programme theory is problematic. And further conceptual confusion arises when the schematic (that is decontextualised!) is identified as “Middle-range programme theory”. Please carefully address this. To me this figure is an attempt at creating a comprehensive mapping of key enablers and mechanisms identified but NOT really either an MRT or a PT. Indeed, upon p.16, their assertions in “How do psychosocial groups lead to positive outcomes” are closer to explanatory MRTs that emerge from this review than this schematic in my assessment (albeit will need more work).

An important gap has been the absence of evidence/insights on key groups that are known to face social exclusion including (but not limited to) caste, religious minorities, LGBTQI, Adivasis/poor migrants/homeless. Without evidence on these, the discussion that the authors bring up on p.14 on “groups are socially acceptable” will need critical re-assessment.

Several points in the discussion veer beyond the remit of their review. Can be potentially reduced in length keeping to the findings in the current review.

Minor/discretionary

- On p.3, see phrase “..which seek provide an explanatory analysis”. Check grammar.

- PRISMA diagram (fig 1) has a remnant of grammar auto-correct while taking a screenshot (probably) which appears over the word “removed”.

- The list of countries covered in the review appears too often (3 times?)

- On p.13, both “collective action” and collective strength are indicated as mechanism 4 at diff points.

- On p.13, need not capitalise O in Figure “One”

Reviewer #2: Thank you for inviting me to peer review the manuscript ‘Addressing mental ill-health through psycho-social groups in South Asia – a scoping review using a realist lens’. I have concerns about the methodological rigor and/or reporting of this work. Please find below a few comments to consider:

Abstract

(1) Form authors’ descriptions, it is not clear (i) which databases (including platforms) were searched (ii) for what type of literature (iii) published in what languages. In general, systematic search methodology for scoping reviews should be explicit, transparent, and reproducible.

(2) What is the justification for the publication date restriction, i.e., 2007 onwards?

(3) Not clear from the authors’ descriptions if grey literature was searched, please expand.

(4) Did the authors develop a protocol for this review? If yes, was it registered with any of the platforms or published? If not, please explain.

(5) Did the authors consider a critical appraisal of the literature given they are looking at intervention research? If yes, what tool did they use?

(6) Did the authors follow the PRISMA extension for scoping reviews?

(7) It will be informative to report here the number and characteristics of participants the included interventions report on.

Methods and materials

(8) I would suggest rewriting the methods sections as it reads like an outline of the protocol.

(9) Page 5- Design/Research type - I am not clear why study protocols were included?

(10) The authors should make it clear to the reader what the inclusion and exclusion criteria were. Did the authors exclude commentaries, books, abstracts, and theoretical studies, opinion pieces, newsletters, magazines, newspapers?

(11) What is the definition of adults in this work? I can see from Supplementary file A that interventions targeting children (14 and 15 years olds) were included, please explain.

(12) Systematic search methodology for scoping reviews should be explicit, transparent, and reproducible. Further, the authors should provide a complete search for at least one database (e.g., MEDLINE) so it is clear for the reader what type of MeSH terms and keywords were used, which terms were exploded, restrictions applied and the number of hits retrieved for each search string. Please add a full search for MEDLINE.

(13) Could the authors explain Table 1?

(i) Where these items are coming from? Why the authors did not use a validated quality assessment tool that would account for all types of studies?

(ii) How the results of the quality assessment exercise have been used in findings.

(iii) What do the authors mean by intervention studies? Why intervention studies are described separately from quantitative studies. RCTs are interventions that report quantitative evidence.

(14) I am not clear why the authors report on Ethical considerations for the scoping review, please explain.

(15) I would urge the authors to conform to the PRISMA extension for scoping review guidelines.

Results/Discussion

(16) From the authors’ descriptions, I am not clear what type of evidence was extracted from what type of studies.

(17) In the peer-review PDF, I cannot see Table 2 but I have access to 2 PRISMA diagrams. I am therefore unable to evaluate if the data and analyses support the results and discussion.

(18) What about the effectiveness of interventions? I cannot see any synthesis and further discussion around effect sizes coming from RCTs, please explain.

(19) I am not clear from the title what Table 3 describes. Are those measures/validated tools used for measuring mental health in interventions? If yes, why social factors are included as an outcome, please expand on this.

(20) In the peer-review PDF, I cannot see Table 4. I am therefore unable to evaluate if the data and analyses support the results and discussion.

(21) I would suggest the review findings are described separately from the engagement work with panelists and other stakeholders.

(22) The manuscript will benefit from revision for typographical errors.

6. PLOS authors have the option to publish the peer review history of their article (what does this mean?). If published, this will include your full peer review and any attached files.

**Do you want your identity to be public for this peer review?** For information about this choice, including consent withdrawal, please see our Privacy Policy.

Reviewer #1: **Yes: **Prashanth N Srinivas

Reviewer #2: No

---

## [Decision Letter · Decision Letter 1]

29 May 2023

PGPH-D-22-00857R1

FULL Improving mental ill-health with psycho-social group interventions in South Asia – a scoping review using a realist lens

Dear Dr. Mathias,

Thank you for submitting your manuscript to PLOS Global Public Health. After careful consideration, we feel that it has merit but does not fully meet PLOS Global Public Health’s publication criteria as it currently stands. Therefore, we invite you to submit a revised version of the manuscript that addresses the points raised during the review process.

We kindly recommend:

Define the term - *psychosocial group intervention* at the beginning of the article,Remove information about the exclusion criteria from the supplementary material to the Methods section,Elaborate the listed limitations and strengths, explaining why they are strengths or limitations,Update the references to the latest Global Burden of Disease Study,Explain all the points mentioned by the Reviewer 3,Review the text for typos and errors, again.

We look forward to receiving your revised manuscript.

Kind regards,

Hanna Nalecz, Ph.D.

Academic Editor

Journal Requirements:

Additional Editor Comments (if provided):

Reviewers' comments:

Reviewer's Responses to Questions

**Comments to the Author**

1. If the authors have adequately addressed your comments raised in a previous round of review and you feel that this manuscript is now acceptable for publication, you may indicate that here to bypass the “Comments to the Author” section, enter your conflict of interest statement in the “Confidential to Editor” section, and submit your "Accept" recommendation.

Reviewer #1: All comments have been addressed

Reviewer #3: (No Response)

2. Does this manuscript meet PLOS Global Public Health’s publication criteria? Is the manuscript technically sound, and do the data support the conclusions? The manuscript must describe methodologically and ethically rigorous research with conclusions that are appropriately drawn based on the data presented.

Reviewer #1: Yes

Reviewer #3: Yes

3. Has the statistical analysis been performed appropriately and rigorously?

Reviewer #1: N/A

Reviewer #3: N/A

4. Have the authors made all data underlying the findings in their manuscript fully available (please refer to the Data Availability Statement at the start of the manuscript PDF file)?

Reviewer #1: Yes

Reviewer #3: Yes

5. Is the manuscript presented in an intelligible fashion and written in standard English?

Reviewer #1: Yes

Reviewer #3: Yes

6. Review Comments to the Author

Reviewer #1: Thank you for the opportunity to read the revised version. I have no further comments to offer on this version.

Reviewer #3: Overall a well-written paper on an important topic. It looks like the authors have addressed previous reviewers' comments well. I would recommend a few additional revisions prior to publication:

Methods

• Confused by connection of search strategy to research questions - Research questions focus on “psycho-social group” interventions but “psycho-social group” and related terms, e.g., “peer support group” were not included in search

• Response “Exclusion criteria are supplied in the supplementary material.” – should be included in main body of paper in Methods section

• Further clarify sentence “It was applied on the level of the intervention, not on the level of individual papers, since we recognise that different analyses are described in varying levels of detail in different publications.”

• Why only meetings with experts in 2 countries when multiple other countries included in South Asia? Why Practitioner and Lived Experience panels only in India?

Discussion

• The issue of measures (p. 25) could be elaborated on, i.e., what is the impact of non-locally developed measures on study findings?

• Methodological Considerations sub-heading could be revised to Strengths and Limitations

- 1st para should be included in methods section; more on study motivation and organization responsible should be included in Intro

- Limitations and strengths should be elaborated on – rather than just listing them, explain why they are strengths or limitations

- Some editorial revisions needed in this section (typos, errors)

Conclusion

• “This is the first review globally to exclusively examine the contribution of psycho-social groups as a mental health intervention.”

- This isn’t true – may be true that it is first review in South Asian region?

- Define “psycho-social group intervention” at start of paper

References

- Update to latest GBD Study (2019)

7. PLOS authors have the option to publish the peer review history of their article (what does this mean?). If published, this will include your full peer review and any attached files.

**Do you want your identity to be public for this peer review?** For information about this choice, including consent withdrawal, please see our Privacy Policy.

Reviewer #1: **Yes: **Prashanth N Srinivas

Reviewer #3: **Yes: **Dr Farah N. Mawani

---

## [Decision Letter · Decision Letter 2]

31 Jul 2023

FULL Improving mental ill-health with psycho-social group interventions in South Asia – a scoping review using a realist lens

PGPH-D-22-00857R2

Dear Dr Mathias,

We are pleased to inform you that your manuscript 'FULL Improving mental ill-health with psycho-social group interventions in South Asia – a scoping review using a realist lens' has been provisionally accepted for publication in PLOS Global Public Health.

Best regards,

Hanna Nalecz, Ph.D.

Academic Editor

Reviewer Comments (if any, and for reference):

Reviewer's Responses to Questions

**Comments to the Author**

1. If the authors have adequately addressed your comments raised in a previous round of review and you feel that this manuscript is now acceptable for publication, you may indicate that here to bypass the “Comments to the Author” section, enter your conflict of interest statement in the “Confidential to Editor” section, and submit your "Accept" recommendation.

Reviewer #1: All comments have been addressed

2. Does this manuscript meet PLOS Global Public Health’s publication criteria? Is the manuscript technically sound, and do the data support the conclusions? The manuscript must describe methodologically and ethically rigorous research with conclusions that are appropriately drawn based on the data presented.

Reviewer #1: Yes

3. Has the statistical analysis been performed appropriately and rigorously?

Reviewer #1: N/A

4. Have the authors made all data underlying the findings in their manuscript fully available (please refer to the Data Availability Statement at the start of the manuscript PDF file)?

Reviewer #1: Yes

5. Is the manuscript presented in an intelligible fashion and written in standard English?

Reviewer #1: Yes

6. Review Comments to the Author

Reviewer #1: All comments have been addressed.

7. PLOS authors have the option to publish the peer review history of their article (what does this mean?). If published, this will include your full peer review and any attached files.

**Do you want your identity to be public for this peer review?** For information about this choice, including consent withdrawal, please see our Privacy Policy.

Reviewer #1: **Yes: **Prashanth N Srinivas
